# Some Constructions and Mathematical Properties of Zero-Correlation-Zone Sonar Sequences

**DOI:** 10.3390/e26040317

**Published:** 2024-04-05

**Authors:** Xiaoxiang Jin, Gangsan Kim, Sangwon Chae, Hong-Yeop Song

**Affiliations:** School of Electrical and Electronic Engineering, Yonsei University, Seoul 03722, Republic of Korea; xxjin@yonsei.ac.kr (X.J.); gs.kim@yonsei.ac.kr (G.K.); sw.chae@yonsei.ac.kr (S.C.)

**Keywords:** sonar sequences, zero-correlation-zone, costas arrays, distinct difference property

## Abstract

In this paper, we propose the zero-correlation-zone (ZCZ) of radius *r* on two-dimensional m×n sonar sequences and define the (m,n,r) ZCZ sonar sequences. We also define some new optimality of an (m,n,r) ZCZ sonar sequence which has the largest *r* for given *m* and *n*. Because of the ZCZ for perfect autocorrelation, we are able to relax the distinct difference property of the conventional sonar sequences, and hence, the autocorrelation of ZCZ sonar sequences outside ZCZ may not be upper bounded by 1. We may sometimes require such an ideal autocorrelation outside ZCZ, and we define ZCZ-DD sonar sequences, indicating that it has an additional distinct difference (DD) property. We first derive an upper bound on the ZCZ radius *r* in terms of *m* and n≥m. We next propose some constructions for (m,n,r) ZCZ sonar sequences, which leads to some very good constructive lower bound on *r*. Furthermore, this construction suggests that for *m* and *r*, the parameter *n* can be as large as possible indefinitely. We present some exhaustive search results on the existence of (m,n,r) ZCZ sonar sequences for some small values of *r*. For ZCZ-DD sonar sequences, we prove that some variations of Costas arrays construct some ZCZ-DD sonar sequences with ZCZ radius r=2. We also provide some exhaustive search results on the existence of (m,n,r) ZCZ-DD sonar sequences. Lots of open problems are listed at the end.

## 1. Introduction

Sonar sequences are two-dimensional synchronizing patterns of dots and blanks with minimal ambiguity [1]. Rectangular array representation of sonar sequences is defined by having only one dot per column and the distinct difference properties of the dots. They are used in active sonar systems to improve target detection performance. They are also useful in radar [2] and many other applications where optimal 2-dimensional autocorrelation is required. Costas arrays as sonar sequences were introduced by J. P. Costas in 1984 [3]. Subsequently, research interest was aroused in the existence [4], enumeration, construction [5,6,7,8], and mathematical properties [9,10,11,12] of Costas arrays and also sonar sequences. It is further generalized into various shapes, for example, honeycomb array, maintaining the distinct difference properties [13].

Numerous studies have delved into the structural properties of sonar sequences. The properties of symmetry were discussed for Welch and Golomb constructions in [14]. S. W. Golomb and H. Taylor had previously proposed a weakened version of the conjecture, asserting that single periodicity was characteristic of Welch construction of the Costas array. Subsequently, in [15], the concept of cyclic Costas sequences was introduced, along with the conjecture that a Costas sequence is cyclic if and only if it is Welch.

With the introduction of Costas arrays, the search for the number of Costas arrays began using computers, even though computers were not well developed at the time. R. Games and M. Chao were the first to report exact values for order n≤10, the values for n≤12 were found by J. P. Costas, and J. Robbins furthered the search up to n=13 in 1984 [8]. In 1988, Silverman et al. reported a further extension to n=17, and developed a probabilistic estimation formula for the number of Costas arrays [16]. In 2011, K. Drakakis and S. Rickard et al. published results enumerating Costas arrays up to n=29 [17].

In one dimension, a binary sequence is called to have an ideal autocorrelation when the out-of-phase autocorrelation magnitude is at most 1 or 2 according to the period of the sequence [18,19]. In [20], P. Fan proposed the concept of zero-correlation-zone (ZCZ) in which the autocorrelation is zero (perfect). As one-dimensional CDMA sequences, binary or non-binary ZCZ sequences can be used to perfectly eliminate co-channel and multipath interference for quasi-synchronous CDMA systems [21]. Therefore, they do not care for the autocorrelation values of ZCZ sequences outside ZCZ. It would have been mathematically or theoretically desirable if the autocorrelation magnitudes of ZCZ sequences were also very low or even close to zero, which is not at all required for QS-CDMA systems using ZCZ sequences since the system performance depends only on the autocorrelation magnitudes inside ZCZ [21,22]. In addition, several studies on the construction and bounds of these one-dimensional ZCZ sequences have been published [22,23,24].

Sonar sequences are two-dimensional synchronizing patterns since their autocorrelation properties are well-described in two dimensions. We now propose for the first time the concept of ZCZ into sonar sequences, and we emphasize that we do care for the autocorrelation value zero inside ZCZ (except for the in-phase, of course) and we do not care for the value outside ZCZ. As we have mentioned in the previous paragraph, it would have been great if we could define a ZCZ sonar sequence that has not only the zero autocorrelation value inside ZCZ but also the value of at most 1 outside ZCZ. We will call such a ZCZ sonar sequence a ZCZ-DD sonar sequence, which is itself a sonar sequence. We note, therefore, that a ZCZ sonar sequence may not be a (conventional) sonar sequence since it may lack the so called distinct difference property that defines sonar sequences.

In this paper, we propose the zero-correlation-zone (ZCZ) of radius *r* on two-dimensional m×n sonar sequences and define the (m,n,r) ZCZ sonar sequences. We also define some new optimality of an (m,n,r) ZCZ sonar sequence which has the largest *r* for given *m* and *n*. Because of the ZCZ for perfect autocorrelation, we are able to relax the distinct difference property of the conventional sonar sequences, and hence, the autocorrelation of ZCZ sonar sequences outside ZCZ may not be upper bounded by 1. We may sometimes require such an ideal autocorrelation outside ZCZ, and we call this a ZCZ-DD sonar sequence, indicating that it has an additional distinct difference (DD) property.

It is to be noted that, for conventional sonar sequences (without considering ZCZ), one has to increase the value *n* for a given *m* in order to increase the overall autocorrelation performance [25,26]. This gives the definition of (conventional) optimal (m,n) sonar sequences with largest *n* for a given *m* [26]. Now, we emphasize that it is quite appropriate to think of the new optimality of (m,n,r) ZCZ sonar sequences with largest value of *r* given *m* and *n*, since now it has perfect autocorrelation inside ZCZ and hence it is desirable to have as large ZCZ as possible for given size m×n. It is because we may be able to limit the operating range (of active sonar systems) inside ZCZ, as is the case for the one-dimensional sequences with ZCZ [20].

In Section 2, we review sonar sequences, encompassing essential definitions, properties, autocorrelation properties, and some well-known constructions. We introduce the Manhattan metric, which will be used in subsequent discussions to represent the ZCZ radius. Section 3 contains some main results on ZCZ sonar sequences. Section 4 discusses some theory only on ZCZ-DD sonar sequences. Section 6 discusses some open problems of both ZCZ sonar sequences and ZCZ-DD sonar sequences.

## 2. Preliminary

**Definition 1.** 
*(Sonar Sequences, Sonar Arrays and Costas Arrays [1,26]) Let m≤n be positive integers. A function f:{1,2,…,n}→{1,2,…,m} has the distinct difference (DD) property if*

f(u+h)−f(u)=f(v+h)−f(v)impliesu=v

*for 1≤h≤n−1 and 1≤u,v≤n−h.*

*An (m,n) sonar sequence is a function f:{1,2,…,n}→{1,2,…,m} with the DD property. It can be written as*

f=[f(1),f(2),…,f(n)],

*where 1≤f(j)≤i, for j=1,2,...,n. This can also be represented as an m×n sonar array A=[A(i,j)], where*

$$A\left(i,j\right)=\left\{\begin{array}{cc} 1, & f(j)=i\\ 0, & otherwise \end{array}\right.,1\leq i\leq m,1\leq j\leq n.$$

*It is a usual convention to represent "1" with a dot and "0" with a blank in A.*

*An (m,n) sonar sequence is called optimal if n is the largest with m rows.*

*The Costas array is a sonar array of square size with an additional condition that there is only one "dot" in each row.*


There are some well-known constructions of Costas arrays as sonar sequences.

**Lempel construction** [1,5,8]: Let q>2 be a prime or a prime power and let α be a primitive element of $F_{q}$ which is the finite field of size *q*. Then f:{1,2,…,q−2}→{1,2,…,q−2} defined by the relation $\alpha^{j}+\alpha^{f(j)}=1$, for j=1,2,...,q−1, is a (q−2)×(q−2) Costas array.**(Exponential) Welch construction** [1,5,8]: Let α be a primitive element of $F_{p}$ where *p* is a prime. Then f:{1,2,…,p−1}→{1,2,…,p−1} defined by $f\left(i\right)=\alpha^{i}$, for i=1,2,...,p−1, is a (p−1)×(p−1) Costas array. Furthermore, if [f(1),f(2),...,f(p−1)] is the exponential Welch Costas array, then so is
[f(j),f(j+1),...,f(p−1),f(1),f(2),...,f(j−1)]
for each j=2,3,...,p−1. This property is called the single periodicity of the Costas array.

There are also Quadratic constructions [25], Shift construction [27], Golomb construction [5,8], the constructions using Sidon set [28] and their extensions.

We note that attaching *i* empty rows, for any integer i=1,2,..., to the above n×n Costas array gives an (n+i)×n sonar array. We note also that rotating the rows of the result any number of times is still an (n+i)×n sonar array. If we start from the exponential Welch Costas array, then rotating the columns of the result any number of times is still an (n+i)×n sonar array because of its single periodicity. This will be used later for the some new construction of ZCZ-DD sonar sequences with radius 2.

Another variation on any n×n Costas array is to delete any corner dot and obtain the (n−1)×(n−1) Costas array. Deleting the corner dots twice gives the size (n−2)×(n−2). Two corner dots in the diagonal position can be deleted once to produce the size (n−2)×(n−2). This will also be used later for ZCZ-DD sonar sequence construction.

The discrete non-periodic autocorrelation function [8] C(τ,φ) of an m×n sonar array [A(i,j)] where i=1,2,…,m, j=1,2,…,n is defined to be the number of coincidences between dots in A(i,j) and its shift A(i+φ,j+τ) where τ is the amount of horizontal shift and φ is the amount of vertical shift. The set of all the values of C(τ,φ) can be represented as an array of size (2m−1)×(2n−1), and it has a center-symmetric structure:$$C\left(\tau ,\varphi \right)=C\left(-\tau ,-\varphi \right),\;\mathrm{for}\;\mathrm{all}\;\left(\tau ,\varphi \right)\in Z^{2}.$$
When (τ,φ)=(0,0), the correlation has the peak value C(0,0)=n, since all the dots coincide. The DD property implies that
C(τ,φ)≤1,for all(τ,φ)≠(0,0).

In this paper, we will define ZCZ in the Manhattan metric. The Manhattan metric is also known as the taxicab metric [29]. In a 2-dimensional plain, the Manhattan distance *D* between two dots in positions $a=(x_{1},y_{1})$ and $b=(x_{2},y_{2})$ is defined by
(1)$$D\left(a,b\right)=|x_{1}-x_{2}\left|+\right|y_{1}-y_{2}|.$$
We will use the terms “sonar array” and “sonar sequence” interchangeably, and hence, the autocorrelation of a sonar sequence has to be understood as that of the sonar array.

This induces an integer-valued lattice in the 2-dimensional plain, which will be denoted by $Z^{2}$. For a positive integer *r*, we consider the Manhattan-circle of radius *r* centered at the origin in $Z^{2}$ and the set M(r) of all the integer points inside, i.e.,
$$M\left(r\right)=\left\{\left(x,y\right)\in Z^{2}:|x|+\left|y\right|\leq r\right\}.$$

**Lemma 1.** 
*The area of a Manhattan-circle M(r) of radius r is given as*

(2)
$$\left|M\left(r\right)\right|=1+2r+2r^{2}.$$



**Proof.** The first term in (Equation 2) counts the center. The remaining size is given by 4 times (1+2+⋯+r) as shown in Figure 1. □

## 3. Main Results on ZCZ Sonar Sequences

**Definition 2.** 
*For positive integers r and m≤n, an (m,n,r) ZCZ sonar sequence is a function f:{1,2,…,n}→{1,2,…,m} such that its autocorrelation C(τ,φ)=0 for all (τ,φ)≠(0,0) with |φ|+|τ|≤r where r is the radius of ZCZ in the Manhattan metric.*


For clarity, it is important to note that a ZCZ sonar sequence, although termed a “sonar sequence”, does not necessarily satisfy the DD property of sonar sequences. A special case of ZCZ sonar sequences is a ZCZ-DD sonar sequence, which is a sonar sequence with ZCZ of some radius.

**Definition 3.** 
*An (m,n,r) ZCZ-DD sonar sequence is an (m,n,r) ZCZ sonar sequence that has the DD property, in addition.*


**Remark 1.** 
*A ZCZ-DD sonar sequence is a sonar sequence. A ZCZ-DD sonar sequence is always a ZCZ sonar sequence, but not conversely. A ZCZ sonar sequence may not have the DD property.*


**Example 1.** 
*The first example in Figure 2 is a (5,5) Costas array (sonar sequence) and its autocorrelation. The second example is a (5,5,2) ZCZ-DD sonar sequence and its autocorrelation showing the ZCZ of radius 2. We note that it is a sonar sequence. The third example is a (5,5,2) ZCZ sonar sequence. Observe that it is not a sonar sequence because it does not have the DD property. This can be seen by the value 2 at some out-of-phase shifts (τ,φ)≠(0,0).*


Given an (m,n,r) ZCZ sonar array, we define $D_{min}$ to be the minimum Manhattan distance among all the distances between the pairs of dots. The following is obvious:

**Lemma 2.** 
*For a ZCZ sonar sequence with its ZCZ radius r, the distance D of any pair of dots satisfies*

$$D\geq D_{min}=r+1.$$



This lemma gives some trivial upper bound on *r* of any (m,n,r) ZCZ sonar sequences. That is,
$$r=D_{min}-1\leq D-1.$$
The following upper bound on *r* is the best in terms of *m* and n≥m that we could prove. The main idea is analogous to the proof of Hamming bound on binary linear block codes:

**Theorem 1.** 
*For an (m,n,r) ZCZ sonar sequence with n≥m≥2, we have*

(3)
$$r\leq \left\lfloor \sqrt{2m-4}\right\rfloor .$$



**Proof.** Consider the m×n array corresponding to an (m,n,r) ZCZ sonar sequence. Then, any two Manhattan-circles of radius ρ with dots at the center will not intersect with each other when $\rho \geq \left\lfloor \frac{r}{2}\right\rfloor$. Therefore, the sum of the area of all the Manhattan-circles of radius ρ cannot be more than the total area of the array. Here, we may have to consider the dots in the edges so that the Manhattan-circle may cover the area beyond the m×n sonar array. For this, we increase the total area of the array from m×n to (m+2ρ)×(n+2ρ), since the dot in the edges could reach the distance ρ for both horizontally and vertically. By carefully counting the number of cells in these additional areas, we see that we only have to increase in one direction. Therefore, we have the bound (similar to Hamming bound in algebraic coding theory)
$$n\left|M(\rho )\right|\leq \left(m+2\rho \right)n,$$
or
$$\left|M(\rho )\right|\leq \left(m+2\rho \right).$$
We substitute ρ=r/2 on LHS and ρ=(r−1)/2 on RHS by carefully counting again the additional areas outside the m×n array. This gives the bound in the theorem. □

In [30], the maximum number of disjoint non-attacking Queens (NAQ) patterns that can sit on an n×n chessboard is proposed, where each pattern consists of *n* NAQs placed symmetrically around the center. NAQ means that for each dot in the pattern, there are no other dots in the horizontal, vertical and diagonal directions. We cite one of NAQ patterns and calculate the autocorrelation as shown in Figure 3. From its autocorrelation, it can be seen that it is an (11,11,2) ZCZ sonar sequence. The following construction of (m,n,r) ZCZ sonar sequence essentially comes from this idea from [30].

**Theorem 2.** 
*The function f:{0,1,…,n−1}→{0,1,…,m−1} defined by f(j)=rj(modm) is an (m,n,r) ZCZ sonar sequence for any positive integers m,n≥m and r≥3 with $m=r^{2}-1$.*


**Proof.** We will focus on a section containing consecutive *r* columns inside. For j=0,1,2,...,r−1 and also j=r, the function *f* is shown in Figure 4. We claim that the Manhattan distance between any two dots is at least r+1 as shown below, and since this one period of *r* columns can repeat indefinitely, the proof is completed:Consider the columns from 0 to r−1 in Figure 4. Any two adjacent dots have Manhattan distance r+1. Consider the dot in column *r* in Figure 4. Since $m=r^{2}-1$, it is located one row up relative to the dot in column 0. Therefore, the Manhattan distance (of lower arrow in the figure) is r+1. The Manhattan distance of the dot from the dot in column 1 is r−1+r−1=2r−2≥r+1 since r≥3 (upper arrow in the figure). □

The above construction gives a constructive lower bound on the parameter *r* for (m,n,r) ZCZ sonar sequences:

**Corollary 1.** 
*There exists an (m,n,r) ZCZ sonar sequence such that*

$$r\geq \sqrt{m+1}.$$



In an attempt to improve the lower bound in the above corollary, we propose another construction for (m,n,r) ZCZ sonar sequences in which the first repeat of the dot in the lowest block of size *r* is not in column *r* (which is the case in Figure 4 for the proof of Theorem 2) but in some column *t* where t<<r where we assume that r≥3. Since the total number of rows is *m*, its row index must be rt(modm) which is equal to rt−m. Therefore, we are trying to find the minimum t∈{0,1,…,n−1} such that rt−m>0, where the function f:{0,1,…,n−1}→{0,1,…,m−1} defined as f(j)=rj(modm) is shown in Figure 5.

Since any two adjacent dots in the first *t* columns have the Manhattan distance of r+1, we need to check only the distances between the dot in column *t* and the dots in the first and second columns. This gives the following inequalities:(4)r+1≤t+rt−m
and
(5)r+1≤t−1+r−(rt−m).

By combining (Equation 4) and (Equation 5), we obtain the following range of *m*:(6)(r−1)t+2≤m≤(r+1)(t−1)
or the inequality
(r−1)t+2≤(r+1)(t−1)
which implies that
(7)$$\frac{r+3}{2}\leq t.$$

Now, a ZCZ sonar sequence can be constructed for (m,n,r) as follows: Choose a positive integer r≥3, and select the parameter *t* satisfying $t\geq \lceil \frac{r+3}{2}\rceil$. The pair of integers *r* and *t* determines the range of *m* by (Equation 6). Select an appropriate value of *m* in this range. Then, for any positive integer *n*, we have an (m,n,r) ZCZ sonar sequence f(j)=rj(modm) for j=1,2,...,n.

As an example, the case where t=r yields
$$r^{2}-r+2\leq m\leq r^{2}-1$$
for r≥3. Taking the value $m=r^{2}-1$ in this range for t=r is exactly the case of Theorem 2. Taking the value $m=r^{2}-r+2$ on the other hand for the same t=r gives another (m,n,r) ZCZ sonar sequence for any positive integer *n*.

For some specific example, we consider r=6. Then t≥5 and 5t+2≤m≤7(t−1). Therefore, we may construct the (m,n,6) ZCZ sonar sequences for any positive integer *n* and the value *m* in the following range:$$\begin{array}{cc} t=5 & \;\to \;27\leq m\leq 28\\ t=6 & \;\to \;32\leq m\leq 35\\ t=7 & \;\to \;37\leq m\leq 42\\ etc. \end{array}$$

We summarize the discussions above as our main construction for a family of (m,n,r) ZCZ sonar sequences:

**Theorem 3** (Main construction for ZCZ sonar sequences). *The function f:{0,1,…,n−1}→{0,1,…,m−1} defined by f(j)≡rj(modm), as shown in Figure 5, is an (m,n,r) ZCZ sonar sequence for any positive integers n≥m and r≥3 where the value m must be in the range (Equation 6) in which t satisfies the inequality (Equation 7).*

**Example 2.** 
*When r=3, the value of m can be 8 according to Theorem 3. Figure 6 depicts the (8,8,3) ZCZ sonar sequence with their autocorrelation on the left side.*

*When r=4, the construction derived from Theorem 3 produces the square array depicted on the right side of the figure with its autocorrelation. It becomes apparent that the top $\frac{r}{2}-1=1$ row of the array does not contain any dots. Thus, by removing this top row, we arrive at the construction for the size 13×14 with r=4.*

*Both examples can be repeated any number of times so that the result becomes either (8,n,3) ZCZ sonar sequence or (13,n,4) ZCZ sonar sequence for any positive integer n.*


By selecting the minimum value of $t=\lceil \frac{r+3}{2}\rceil$ for r≥3 from the above construction, we derive the minimum value of *m* and hence the best constructive lower bound on *r*:

**Corollary 2.** 
*The minimum value of m in Theorem 3 becomes the following*

$$m=\left\{\begin{array}{cc} \;\frac{r^{2}+2r+1}{2}, & r\;is\;odd\\ \frac{r^{2}+2r+2}{2}, & r\;is\;even \end{array}\right.,$$

*for the value $t=\lceil \frac{r+3}{2}\rceil$. This gives a constructive lower bound as follows: for any positive integer m≥3, there exists an (m,n,r) ZCZ sonar sequence (for any n≥m) with the value r satisfying*

(8)
$$r\geq \sqrt{2m-1}-1.$$



**Proof.** The case of an odd *r* is obvious. When *r* is even, the theorem says the minimum value of $m=\frac{r^{2}+3r}{2}$, and the construction gives an m×n ZCZ sonar array for any n≥m where the top r/2−1 rows are empty. These rows can be further removed to make *m* smaller. The resulting value of *m* becomes $\frac{r^{2}+2r+2}{2}$. □

We show an example of a (61,52,10) ZCZ sonar sequence in Figure 7 found by computer. This is an interesting example since the construction in Theorem 3 for r=10 gives a ZCZ sonar sequence with the smallest value m=61. It is also very special in that it has a periodic structure of a period of 13 columns repeating 4 times. We note that this period can be repeated any number of times to make a (61,13a+b,10) ZCZ sonar sequence for any integers *a* and *b*. Essentially, it gives a family of examples of (61,n,10) ZCZ sonar sequences for any positive integer n≥1.

**Remark 2.** 
*It is obvious that one can find a family of (m,n,r) ZCZ sonar sequences for some given values of m and r with infinitely many values of n. Some evidence we discussed so far can be summarized as follows:*

*1.* 
*The example of (11,11,2) ZCZ sonar sequence in Figure 3 from [30] can be repeated any number of times and the result can be an (11,n,2) ZCZ sonar sequence for any positive integer n.*
*2.* 
*The example of (61,52,10) ZCZ sonar sequence in Figure 7 can be repeated any number of times and the result can be an (61,n,10) ZCZ sonar sequence for any positive integer n.*
*3.* 
*Theorem 2 gives a family of (m,n,r) ZCZ sonar sequences for any r≥3 and $m=r^{2}-1$ but with infinitely many values of the positive integer n.*
*4.* 
*Theorem 3 generalizes Theorem 2. Corollary 2 gives one specific case for m and r with any positive integer n, which is different from those by Theorem 2. Two examples from this construction are shown in Example 2.*


*Therefore, it becomes meaningless to talk about the ‘optimal’ (m,n,r) ZCZ sonar sequence with the maximum value of n for given m and r. Instead, we may define the optimality of an (m,n,r) ZCZ sonar sequence if it has the maximum r for given m and n.*


**Definition 4.** 
*The ZCZ sonar sequence with the maximum r is called optimal for given m and n. In other words, an (m,n,r) ZCZ sonar sequence is optimal when there does not exist an (m,n,r+1) ZCZ sonar sequence.*


We have searched by computer for the true maximum *r* in (m,n,r) ZCZ sonar sequences for *m* up to 78 and *n* in the range from *m* to m+2. We show this result in Table 1. The value *r* in this table is the maximum in the sense of Definition 4. This has been checked exhaustively. Therefore, they all are optimal ZCZ sonar sequences for given *m* and *n*. It is to be noted further that the upper bound in Theorem 1 is not tight since there are cases where this value is not attained. However, we argue that it is quite good since some other many times, this bound or one less value is attained.

We also show the upper bound on *r* from Theorem 1 for comparison. Therefore, any max value in the table must be equal to or smaller than this upper bound. We also show the constructive lower bound from Corollary 2 as well in the last column. As *n* increases from *m*, the max *r* will be non-increasing. When it reaches the lower bound, it will stay forever as *n* increases indefinitely. Therefore, it is enough to show the values of *n* in the range m≤n≤m+2 for 3≤m≤78.

In this range of values of *m*, we see that the difference between the upper bound and the constructive lower bound is either 0 or 1. When they are the same, the max *r* is this value for any n≥m. When they differ by 1, then the max *r* starts either from the upper bound and decreases by 1 somewhere and stays forever or from the lower bound and stays forever. The example of the former case is when m=17 and those of the latter is when m=20.

For example, for m=17, the max *r* for n=17 is 5 which is the upper bound. Since the constructive lower bound is 4 for m=17 and this value is reached at n=18=m+1, we know that the max r=4 stays the same as *n* increases from 18 indefinitely. For the case m=20, the max *r* at n=20 is 5 which is already equal to the lower bound. Therefore, the max r=5 for *n* stays the same as *n* increases indefinitely. We show the three ZCZ sonar sequences with parameters (17,17,5), (17,18,4) and (20,20,5) as follows:$$\begin{array}{cc} (17,17,5): & \;[5,10,15,1,7,12,17,4,9,14,1,6,11,16,3,8,13]\\ (17,18,4): & \;[1,5,9,13,2,6,10,14,1,5,9,13,2,6,10,14,1,17]\\ (20,20,5): & \;[7,12,1,16,5,10,19,2,7,12,17,4,9,14,1,6,11,16,3,20] \end{array}$$
These are shown in Figure 8. These are examples of optimal ZCZ sonar sequences.

## 4. Two Constructions for Zcz-Dd Sonar Sequences with R=2

**Theorem 4.** 
*Let q be a prime or a prime power and α be a primitive element of $F_{q}$ which is the finite field of size q. Consider the Lempel Costas array (j,f(j)) for j=1,2,…,q−2 given by $\alpha^{j}+\alpha^{f(j)}=1$. If α satisfies $\alpha^{2}+\alpha =1$, then deleting the two corner dots at (1,2) and (2,1) gives a (q−4,q−4,2) ZCZ-DD sonar sequence.*


**Proof.** The Lempel Costas array has only one dot in each row and column and is symmetric along the main diagonal [8]. Therefore, there are two types of dot pairs with a Manhattan distance of 2, as shown in Figure 9. One type consists of two consecutive dots along the diagonal (white dot pair), while the other type consists of two adjacent dots on either side of the diagonal (black dot pair).We now claim that the pair of white dots do not exist in Lempel construction for any *q*. When *q* is even, $\alpha^{j}+\alpha^{j}=0\neq 1$ for all *j*. Therefore, no dot may come on the diagonal. In odd *q*, $\alpha^{j}+\alpha^{j}$ are all distinct for j=1,2,…,q−2, with a unique *j* satisfying $\alpha^{j}+\alpha^{j}=1$. Therefore, there exists only one dot on the diagonal. Consequently, only black dot pair type exists in the array and no white dot pairs.When $\alpha^{2}+\alpha =1$, the dots at positions (2,1) and (1,2) constitute the type of black dot pair. Due to the DD property, this is the only dot pair of this type. Thus, removing the dots at positions (2,1) and (1,2) ensures that the Manhattan distance between the remaining pair of dots is at least 3 and hence r=2. The remaining array is a Costas array and maintains the DD property. □

**Example 3.** 
*Figure 10 shows a (7,7) Lempel Costas array from q=9. Deleting the dots (1,2) and (2,1) gives a (5,5,2) ZCZ-DD sonar sequence.*


For the second type construction of ZCZ-DD sonar sequence with r=2 from the exponential Welch construction for Costas arrays, we observe the following:

**Lemma 3.** 
*Let p be a prime and α be a primitive root mod p. There must exist unique j(modp−1) satisfying*

$$\alpha^{j}-\alpha^{j-1}=1\;\mathrm{or}\;-1\;\left(\mathrm{mod}\;p\right).$$



**Proof.** Consider the case $\alpha^{j}-\alpha^{j-1}=1\;\left(\mathrm{mod}\;p\right)$ or $\alpha^{j-1}\left(\alpha -1\right)=1\;\left(\mathrm{mod}\;p\right).$ Since any non-zero element is some power of α, we let $\alpha -1=\alpha^{u}$ for some *u* with 2≤u≤p−2. This gives
$$\alpha^{j-1+u}=1\;\left(\mathrm{mod}\;p\right).$$
Then j−1+u=0(modp−1) or j=p−u is the unique integer mod p−1. The other case is similar. □

**Theorem 5.** 
*Consider the exponential Welch Costas array $f\left(j\right)=\alpha^{j}$ for any consecutive p−1 values of j. Let $j_{0}\;\left(\mathrm{mod}\;p-1\right)$ satisfy $\alpha^{j_{0}}-\alpha^{j_{0}-1}=1\;\left(\mathrm{mod}\;p\right)$ and $i_{0}\;\left(\mathrm{mod}\;p-1\right)$ satisfy $\alpha^{i_{0}}-\alpha^{i_{0}-1}=-1\;\left(\mathrm{mod}\;p\right)$. Then*

$$f\left(j\right)=\alpha^{j}-1-\alpha^{i_{0}}\;\left(\mathrm{mod}\;p\right)$$

*for $j=j_{0},j_{0}+1,\ldots ,j_{0}+p-2$ is a (p,p−1,2) ZCZ-DD sonar sequence.*


**Proof.** The Welch construction satisfies the DD property. According to Lemma 3, since there is a unique $j_{0}$ that satisfies $\alpha^{j_{0}}-\alpha^{j_{0}-1}=1\;\left(\mathrm{mod}\;p\right)$, take *j* when starting from $j_{0}$, the rest of the dots are all at a Manhattan distance greater than 2 from the point $\left(j_{0},\alpha^{j_{0}}\right)$. And of the rest of the dots, only the Manhattan distance between the dot pair $\left(i_{0},\alpha^{i_{0}}\right)$ and $\left(i_{0}-1,\alpha^{i_{0}-1}\right)$ is 2 since $\alpha^{i_{0}}-\alpha^{i_{0}-1}=-1\;\left(\mathrm{mod}\;p\right)$. By adding an empty row at the bottom of the array, called the 0-th row, and cyclically rotating the rows of the array, once the dot $\left(i_{0},\alpha^{i_{0}}\right)$ moves to the top row of the array, then the pairs of dots with a Manhattan distance of 2 are avoided. Consequently, the Manhattan distance between the dots becomes at least 3. After adding a row, the array consists of a total of *p* rows, it retains the DD property because of its single periodicity. □

**Example 4.** 
*There exists a unique integer $i_{0}\;\left(\mathrm{mod}\;p-1\right)$ satisfying $\alpha^{i_{0}}-\alpha^{i_{0}-1}=-1\;\left(\mathrm{mod}\;p\right)$ from Lemma 3. Similarly, there exists a unique integer $j_{0}\;\left(\mathrm{mod}\;p-1\right)$ satisfying $\alpha^{j_{0}}-\alpha^{j_{0}-1}=1\;\left(\mathrm{mod}\;p\right)$. As shown in Figure 11, when p=7, we have α=3, and hence, $i_{0}=2$ and $j_{0}=5$ both mod 6. Therefore, we take the exponential Welch Costas array given as $f\left(j\right)=3^{j}\;\left(\mathrm{mod}\;7\right)$ for $j=j_{0}=5,6,...,10$. This 6×6 array has two adjacent dots in some two consecutive columns whose row indices are $\alpha^{i_{0}-1}=3$ and $\alpha^{i_{0}}=2$ both mod 7. We will make this a 7×6 sonar array by adjoining an empty row at the bottom. Now, rotating all p=7 rows downward $1+\alpha^{i_{0}}=3$ times will place the dot in column $i_{0}$ at the top. The resulting 7×6 array becomes a (7,6,2) ZCZ-DD sonar sequence.*


**Remark 3.** 
*The upper bound on r for (m,n,r) ZCZ sonar sequences can be an upper bound on r for (m,n,r) ZCZ-DD sonar sequences, only because any (m,n,r) ZCZ-DD sonar sequence is an (m,n,r) ZCZ sonar sequence. We expect that this upper bound must be quite loose.*


Some search results for the max *r* in (m,n,r) ZCZ-DD sequences are documented initially in [31] for m≤17 and we extend the search for m≤20 and show the results in Table 2. Where the parameters m=10, n=m+4, and r=2 represent the existence of an optimal (10,14,2) ZCZ-DD sonar sequence. It also implies that there does not exist a (10,14,3) ZCZ-DD sonar sequence.

## 5. Some Relations with Results in [26,30]

This subsection is newly added as a result of some analysis from the comments of the initial reviewers of this manuscript. All of the authors would like to express sincere appreciation for these comments. We have investigated most of the results in both [26] and [30], with emphasis on some possibility of having ZCZ sonar sequences from theirs. Following is some conclusion from this analysis.

Most of the best known m×n sonar sequences in [26] for *m* up to 100 turned out to have no ZCZ at all. We show only two cases here for m=10 and m=30 from [26]. These are 10×16 and 30×37 sonar sequences as shown in Figure 12. These have the largest value of *n* for the given value of m=10 and m=30. The fact that they do not have ZCZ can be seen easily by observing that there exist two adjacent dots of (Manhattan) distance 1.

The main topic of [30] is to find the maximum number of disjoint nonattacking-*n*-queen patterns that simultaneously pack the n×n board. We find one of interesting relation there when n>7 is an odd prime. One of the solution in this case gives not only an (n,n,3) ZCZ sonar sequence, but a family of disjoint *n* ZCZ sonar sequences (which are also *n* disjoint nonattacking-*n*-queen patterns) of the same parameters that simultaneously pack n×n×n cube without attacking each other in three dimensional space as *n*-queen patterns. We may formulate a theorem from this construction as follows. We skip the proof which is quite straightforward. The first conclusion is from [30]. The second conclusion is the relation with ZCZ sonar sequences.

**Theorem 6.** 
*Let n>7 be a prime. Construct the n×n matrix $Q=(q_{i,j})$ with integers mod n for i,j=0,1,...,n−1 as follows:*

*Put $q_{0,j}=1$ for j=(n−1)/2.*

*Put $q_{0,j+2}=q_{0,j}+1\;\left(\mathrm{mod}\;n\right)$ where the subscript j+2 is computed mod n, for j=(n−1)/2,(n−1)/2+2,....*

*For each j=0,1,...,n−1, put $q_{i+1,j}=q_{i,j}+2\;\left(\mathrm{mod}\;n\right)$ where the subscript i+1 is computed mod n for i=0,1,....*

*Then, the first conclusion from [30] is that Q is a packing of an n×n board by n disjoint nonattacking-n-queen patterns, which is three-dimensionally nonattacking queens also. Second (new) conclusion: for each symbol k=0,1,...,n−1, the pattern of the constant symbol k in Q is an (n,n,3) ZCZ sonar sequence.*


## 6. Concluding Remarks

Some immediate open problems on (m,n,r) ZCZ sonar sequences are the following:Describe the values of *m* for which the upper bound on *r* is the same as its constructive lower bound. Some of the smaller such values of *m* from Table 1 are m=5, m=9, m=13, m=14, etc.Describe the values of *m* for which the upper bound on *r* is one more than its constructive lower bound. Some examples of such values of *m* from Table 1 are m=6, m=10, m=15, m=16, etc.Prove that the difference between the upper bound and the constructive lower bound is at most 1 for all positive integers n≥m or else find the values of *m* for which the difference is more than 1.Find the formula for the max *r* for the optimal m×m ZCZ sonar sequence.Prove that the max *r* as n=m,m+1,m+2,... is non-increasing. We know that it eventually reaches and stays at the constructive lower bound in Cor.2.Find any new construction for ZCZ sonar sequences (j,f(j)) for j=1,2,...,n which is not of the type f(j)=rj(modm). Note that the construction in Theorem 6 is of the form f(j)=rj(modn) where r=−4 for all *n* disjoint patterns with some appropriate initial condition. See Figure 13.

For (m,n,r) ZCZ-DD sonar sequences, we have a lot of open problems. Only some of them are listed here:Find the max *r* for given *m* and *n*.Prove that the max *r* as n=m,m+1,m+2,... is non-increasing.Find the max *n* for given *m* and *r*.Find the max n≥m such that r=2 for a given *m*. Some small cases are n=m=5, n=m+1=7, n=m+2=9, and n=m+2=10, etc.Find the relation of *n* and *r* for a given *m*.Find a systematic construction for (m,n,r) ZCZ-DD sonar sequences for r>2.Improve the upper bound on *r* in Remark 3 for given *m* and *n*.

## Figures and Tables

**Figure 1 entropy-26-00317-f001:**
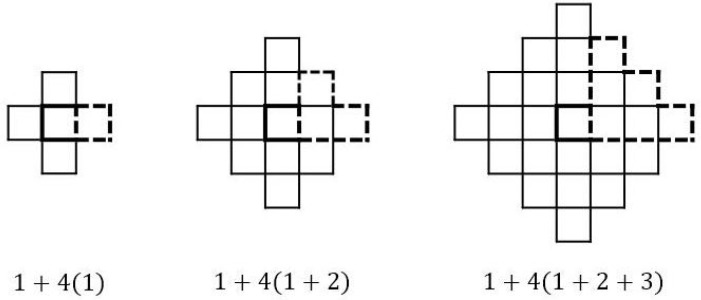
Manhattan-circles of radius r=1,2,3 and its area.

**Figure 2 entropy-26-00317-f002:**
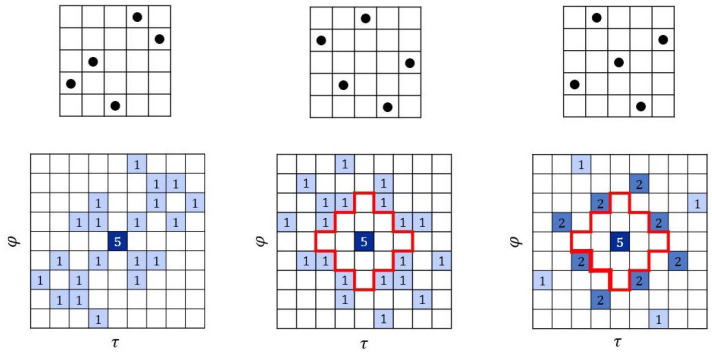
Three 5×5 arrays in Example 1.

**Figure 3 entropy-26-00317-f003:**
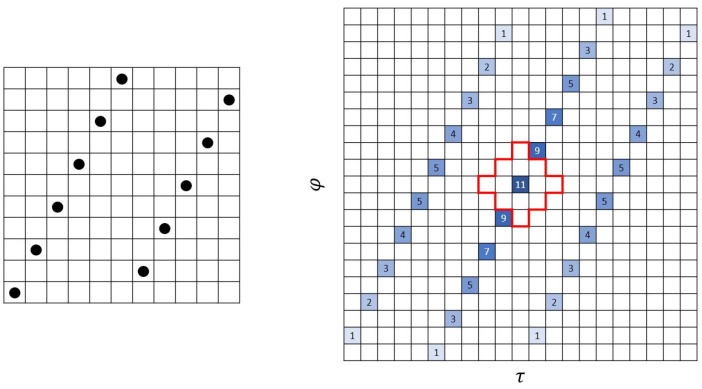
An 11×11 NAQ pattern from [30] is an (11,11,2) ZCZ sonar sequence.

**Figure 4 entropy-26-00317-f004:**
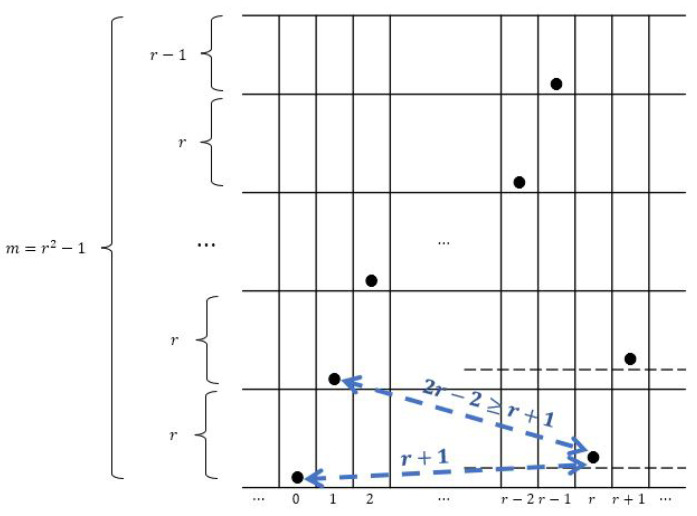
Proof of Theorem 2.

**Figure 5 entropy-26-00317-f005:**
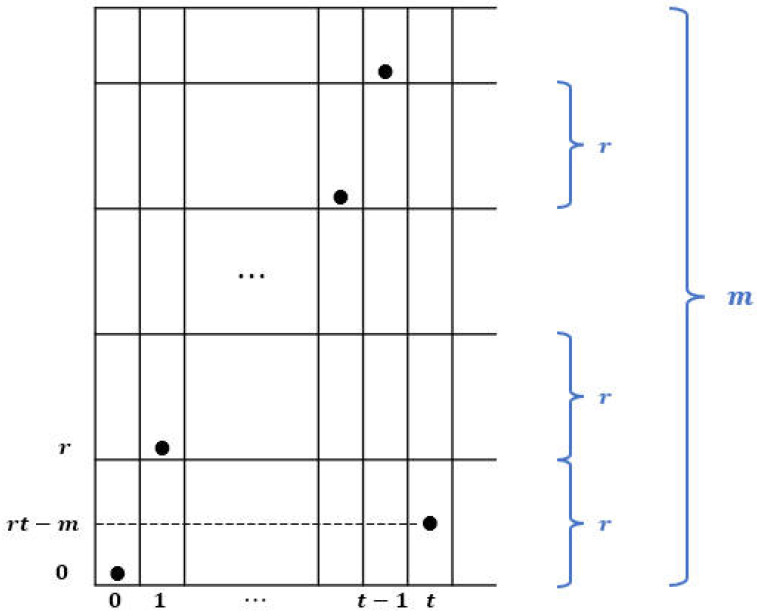
Proof of Theorem 3 where $\frac{\left(r+3\right)}{2}\leq t$ and (r−1)t+2≤m≤(r+1)(t−1).

**Figure 6 entropy-26-00317-f006:**
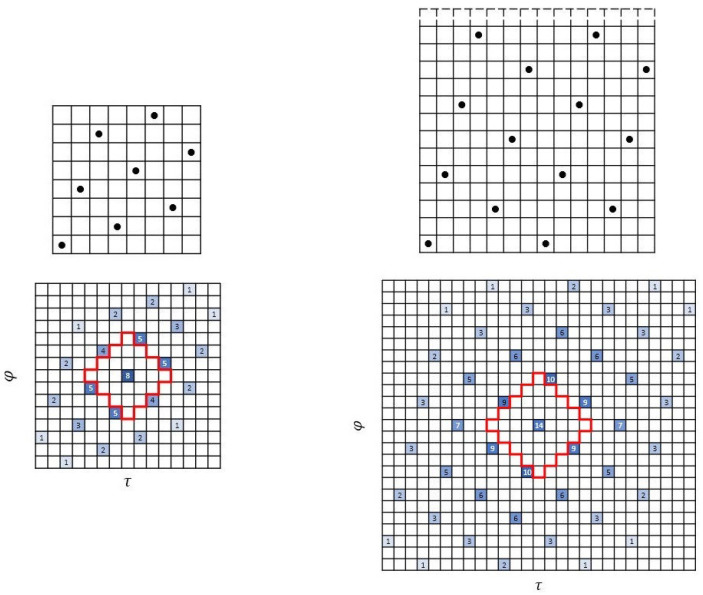
The (8,8,3) and (13,14,4) ZCZ sonar sequences from the construction in Theorem 3.

**Figure 7 entropy-26-00317-f007:**
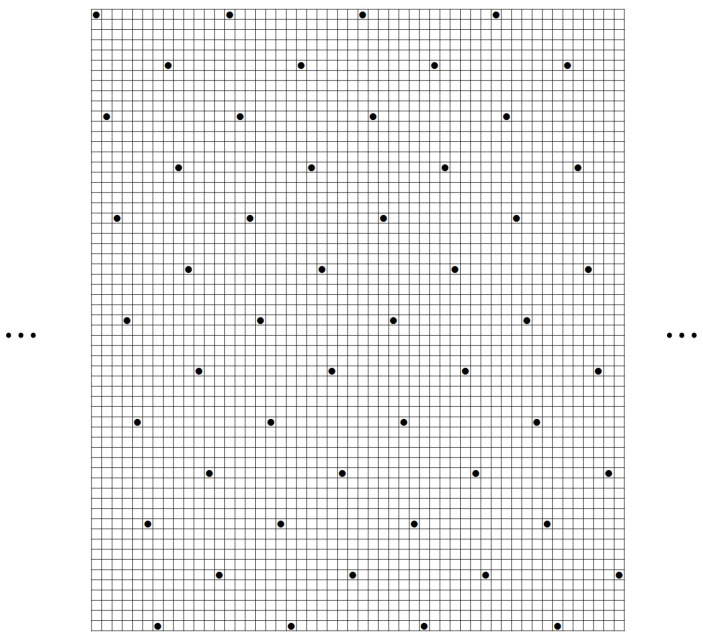
A (61,52,10) ZCZ sonar sequence by computer search.

**Figure 8 entropy-26-00317-f008:**
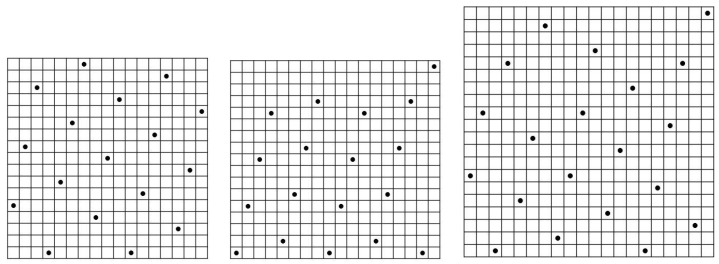
The optimal (17,17,5), (17,18,4) and (20,20,5) ZCZ sonar sequences by computer search.

**Figure 9 entropy-26-00317-f009:**
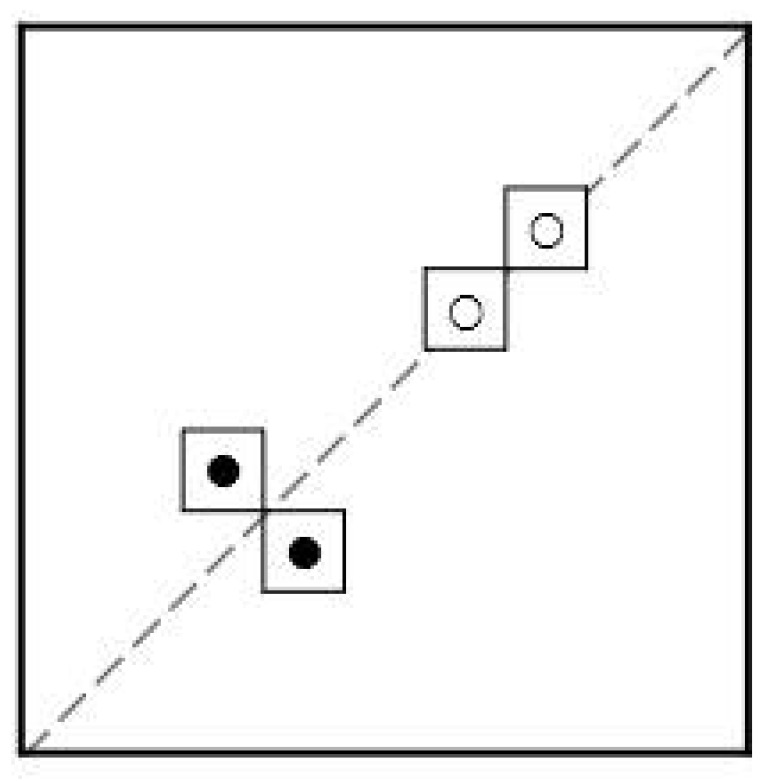
Dot pair types with Manhattan distance 2 in the Lempel construction.

**Figure 10 entropy-26-00317-f010:**
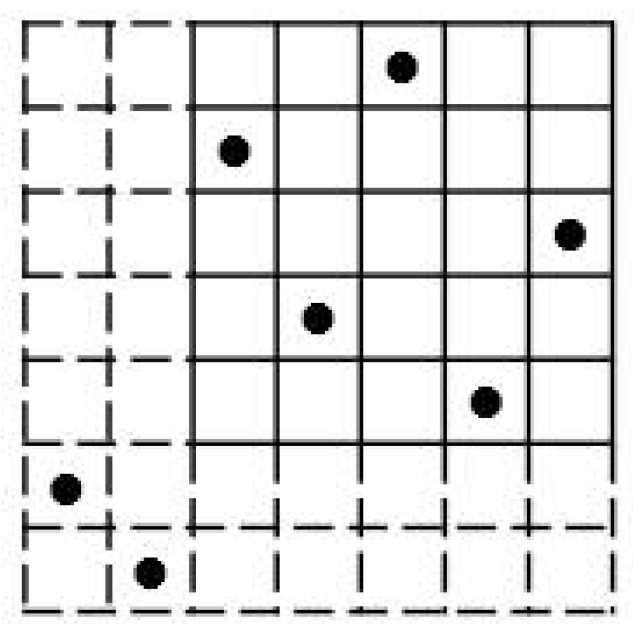
The (5,5,2) ZCZ-DD sonar sequence from the construction in Theorem 4.

**Figure 11 entropy-26-00317-f011:**
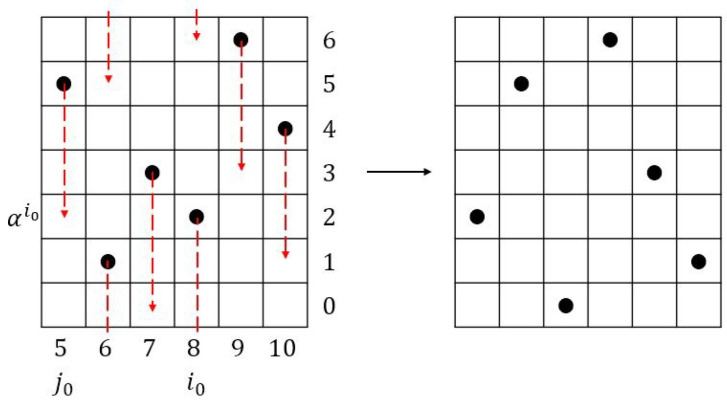
The (7,6,2) ZCZ-DD sonar sequence from the construction in Theorem 5.

**Figure 12 entropy-26-00317-f012:**
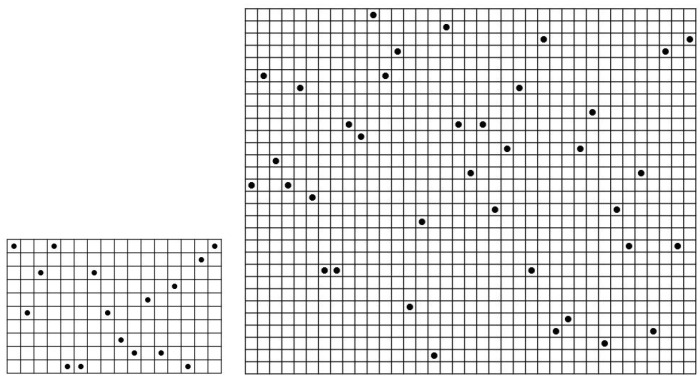
Best known sonar sequences from [26] without ZCZ (m=10 and m=30).

**Figure 13 entropy-26-00317-f013:**
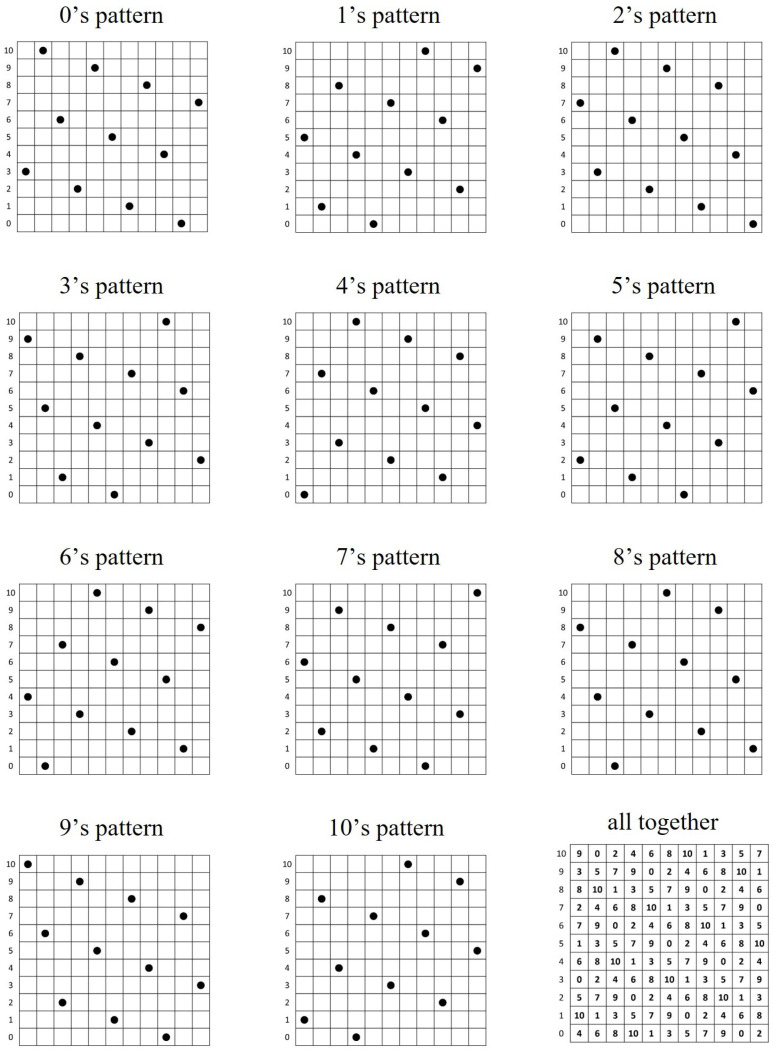
Disjoint nonattacking-*n*-queen patterns (n=11) where each pattern of the constant symbol is an (n,n,3) ZCZ sonar sequence.

**Table 1 entropy-26-00317-t001:** The maximum *r* in (m,n,r) ZCZ sonar sequence found by computer.

	*n*	*m*	m+1	m+2	u.bnd (Theorem 1)	l.bnd (Corollary 2)			*n*	*m*	m+1	m+2	u.bnd (Theorem 1)	l.bnd (Corollary 2)
*m*								*m*						
3		1	1	1	1	1		41		8	8	8	8	8
4		2	1	1	2	1		42		8	8	8	8	8
5		2	2	2	2	2		43		8	8	8	9	8
6		2	2	2	3	2		44		8	8	8	9	8
7		3	2	2	3	2		45		8	8	8	9	8
8		3	3	3	3	2		46		8	8	8	9	8
9		3	3	3	3	3		47		8	8	8	9	8
10		3	3	3	4	3		48		8	8	8	9	8
11		3	3	3	4	3		49		9	9	9	9	8
12		4	3	3	4	3		50		9	9	9	9	8
13		4	4	4	4	4		51		9	9	9	9	9
14		4	4	4	4	4		52		9	9	9	10	9
15		4	4	4	5	4		53		9	9	9	10	9
16		4	4	4	5	4		54		9	9	9	10	9
17		5	4	4	5	4		55		9	9	9	10	9
18		5	5	5	5	4		56		9	9	9	10	9
19		5	5	5	5	5		57		9	9	9	10	9
20		5	5	5	6	5		58		9	9	9	10	9
21		5	5	5	6	5		59		9	9	9	10	9
22		5	5	5	6	5		60		10	10	9	10	9
23		5	5	5	6	5		61		10	10	10	10	10
24		6	5	5	6	5		62		10	10	10	10	10
25		6	6	6	6	6		63		10	10	10	11	10
26		6	6	6	6	6		64		10	10	10	11	10
27		6	6	6	7	6		65		10	10	10	11	10
28		6	6	6	7	6		66		10	10	10	11	10
29		6	6	6	7	6		67		10	10	10	11	10
30		6	6	6	7	6		68		10	10	10	11	10
31		7	6	6	7	6		69		10	10	10	11	10
32		7	7	7	7	6		70		10	10	10	11	10
33		7	7	7	7	7		71		11	11	11	11	10
34		7	7	7	8	7		72		11	11	11	11	10
35		7	7	7	8	7		73		11	11	11	11	11
36		7	7	7	8	7		74		11	11	11	12	11
37		7	7	7	8	7		75		11	11	11	12	11
38		7	7	7	8	7		76		11	11	11	12	11
39		7	7	7	8	7		77		11	11	11	12	11
40		8	8	7	8	7		78		11	11	11	12	11

**Table 2 entropy-26-00317-t002:** The maximum *r* in (m,n,r) ZCZ-DD sonar sequence.

	*n*	*m*	m+1	m+2	m+3	m+4
*m*						
2		1	1	0	−	−
3		1	1	1	0	−
4		1	1	1	1	0
5		2	1	1	1	1
6		2	2	1	1	1
7		2	2	2	1	1
8		2	2	2	1	1
9		2	2	2	1	1
10		2	2	2	2	2
11		2	2	2	2	2
12		3	3	2	2	2
13		3	3	2	2	2
14		3	3	3	2	2
15		3	3	3	3	2
16		3	3	3	3	2
17		3	3	3	3	2
18		3	3	3	3	2
19		3	3	3	3	2
20		3	3	3	3	**?**

## Data Availability

Data is contained within the article.

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
