# Peer review of "Some Constructions and Mathematical Properties of Zero-Correlation-Zone Sonar Sequences"

_entropy, 2024, doi:10.3390/e26040317_

Round 1
Reviewer 1 Report
Comments and Suggestions for Authors
This paper proposed the zero-correlation-zone (ZCZ) of radius r on two-dimensional m × n sonar sequences and define the (m, n, r) ZCZ sonar sequences, along with some constructions and property analysis. The presented concepts and results are original, interesting and significant, leaving lots of potentials for further research.
Unfortunately, most of the basic results (kernels) are obtained mainly by computer search for the true maximum r in (m, n, r) ZCZ sonar sequences for m up to 78 and n in the range from m to m + 2. It would be very good if the existence conditions of (m, n, r) ZCZ-DD sonar sequences and construction mechanism behind these basic results could be disclosed, and extended to arbitrary m. However, this could be the future work as it is quite difficult to do so.
If possible, the author may propose a few conjectures based on the analysis of the searched results.
Reviewer 2 Report
Comments and Suggestions for Authors
Please, compare the advantages of your proposed ZCZ sonar sequences to other sequences published in literature( Costas, reference [25], reference [31], or any other).
As suplemmentary material, it would be interesting that you provide a routine (Matlab, Phyton, ...) to generate an example of your ZCZ sonar sequences. Or indicate another data repository to find such routine.
